# Inhibition of Checkpoint Kinase 1 (CHK1) Upregulates Interferon Regulatory Factor 1 (IRF1) to Promote Apoptosis and Activate Anti-Tumor Immunity via MICA in Hepatocellular Carcinoma (HCC)

**DOI:** 10.3390/cancers15030850

**Published:** 2023-01-30

**Authors:** Xicai Li, Jingquan Huang, Qiulin Wu, Qiang Du, Yingyu Wang, Yubin Huang, Xiaoyong Cai, David A. Geller, Yihe Yan

**Affiliations:** 1Department of General Surgery, The Second Affiliated Hospital of Guangxi Medical University, Nanning 530007, China; 2Thomas E. Starzl Transplantation Institute, Department of Surgery, University of Pittsburgh Medical Center, Pittsburgh, PA 15260, USA

**Keywords:** CHK1, IRF1, DDR, MICA, HCC

## Abstract

**Simple Summary:**

The communication between the DNA damage response pathway and the tumor microenvironment of hepatocellular carcinoma (HCC) has attracted more attention. In this study, high CHK1 expression in HCC tissue was associated with advanced tumor stage and poor prognosis in patients. CHK1 inhibition and cisplatin induced cellular apoptosis via DNA damage in human HCC cells. CHK1 directly bound IRF1 and exerted a proteolytic effect in HCC cells induced by DNA damage. DNA damage induced MICA expression via IRF1 at the transcriptional level in HCC cells. MICA expression was positively correlated with NK cells and CD8+T cell infiltration in human HCC. This study provided new insight into the molecular mechanisms regulating HCC signaling with cross-talk between the IRF1 and CHK1 pathways.

**Abstract:**

Background: CHK1 is considered a key cell cycle checkpoint kinase in DNA damage response (DDR) pathway to communicate with several signaling pathways involved in the tumor microenvironment (TME) in numerous cancers. However, the mechanism of CHK1 signaling regulating TME in hepatocellular carcinoma (HCC) remains unclear. Methods: CHK1 expression in HCC tissue was determined by IHC staining assay. DNA damage and apoptosis in HCC cells induced by cisplatin or CHK1 inhibition were detected by WB and flow cytometry. The interaction of CHK1 and IRF1 was analyzed by single-cell RNA-sequence, WB, and immunoprecipitation assay. The mechanism of IRF1 regulating MICA was investigated by ChIP-qPCR. Results: CHK1 expression is upregulated in human HCC tumors compared to the background liver. High CHK1 mRNA level predicts advanced tumor stage and worse prognosis. Cisplatin and CHK1 inhibition augment cellular DNA damage and apoptosis. Overexpressed CHK1 suppresses IRF1 expression through proteolysis. Furthermore, single-cell RNA-sequence analyses confirmed that MICA expression positively correlated with IRF1 in HCC cells. Immunoprecipitation assay showed the binding between CHK1 and IRF1. Cisplatin and CHK1 inhibition upregulate MICA expression through IRF1-mediated transcriptional effects. A novel specific cis-acting IRF response element was identified at -1756 bp in the MICA promoter region that bound IRF1 to induce MICA gene transcription. MICA may increase NK cell and CD8+T cell infiltration in HCC. Conclusions: DNA damage regulates the interaction of CHK1 and IRF1 to activate anti-tumor immunity via the IRF1-MICA pathway in HCC.

## 1. Introduction

Hepatocellular carcinoma (HCC) is the fifth most common cancer in the world, with high mortality [1]. At present, liver resection and liver transplantation are the main curative options for HCC. Unfortunately, most patients diagnosed with HCC are in the advanced stage, and surgical treatment is not effective. The combination of molecular target therapy with an immune checkpoint inhibitor (ICI) has been used for non-surgical patients. However, the therapeutic resistance and associated adverse reactions limit its application in HCC [2,3,4]. HCC intrinsic and extrinsic factors contributing to therapeutic resistance and adverse reactions contain alterations of several signaling pathways, lack of tumor antigen expression, inhibition of anti-tumor immunity involved in an immune suppressive cell population, as well as cytokine and chemokine release in the tumor microenvironment (TME) [5,6]. Therefore, it is urgent to understand the cellular and molecular mechanisms involved in TME.

Checkpoint kinase 1 (CHK1) plays an important role in the DNA damage response (DDR) pathway and functions as maintaining chromosome integrity and genome stability [7]. CHK1 can resist cellular DNA damage caused by ultraviolet irradiation, ionizing radiation, and chemotherapeutic reagents because it makes cell cycle arrest to repair damaged DNA and promotes cell proliferation as well as exerts anti-apoptotic effects on tumor cells [8]. Some studies confirmed that DDR induced by CHK1 inhibition activates anti-tumor immunity through communicating with several signaling pathways (STAT1-IRF1, cGAS-SITNG) [9,10,11].

Interferon regulatory factor 1 (IRF1) is the first identified member of the IRF families and functions as a central transcriptional factor in the interferon (IFN) pathway characterized by regulating cell proliferation, apoptosis, and tumorigenesis as well as TME [12,13]. Our previous studies demonstrated that IRF1 promoted cellular apoptosis and regulated the recruitment and activation of anti-tumor immune cells in murine liver cancer [14]. Meanwhile, we found that IRF1 suppresses CHK1 expression via microRNA-195 (miR-195) at the post-transcriptional level in HCC [15]. Interestingly, another study showed that CHK1 promoted substrate proteolysis in HCC cells [16]. Thus, whether CHK1 exerts a proteolytic effect on IRF1 and the mechanism of CHK1 inhibition activating immune cells needs further research. 

The nature killer group 2D (NKG2D) receptor is a stimulatory receptor expressed by activated NK cells and CD8+T cells in humans. The related NKG2D ligands contain major histocompatibility complex (MHC) class I associated chains A and B (MICA, MICB) as well as unique long 16 (UL16) binding protein (ULBP) families, which are expressed on human cancer cells. NKG2D ligand expression can be upregulated by DNA damage stress response [17,18] and mediate tumor elimination by cytotoxic lymphocytes through NKG2D receptors [19,20]. Combining anti-MICA/B antibodies and immune modulators can activate NKG2D-dependent NK cells and CD8+T cell response in cancers, including HCC. Conversely, the shedding of MICA/B on the membrane of cancer cells may yield soluble molecules that bind to and downregulate NKG2D on NK cells and CD8+T cells to impair NKG2D-mediated immune responses. Previous studies showed that NKG2D ligand expression was associated with HCC aggressiveness and post-operative recurrence [21,22]. However, the role of NKG2D ligands in the HCC immune response induced by DNA damage has not been fully elucidated.

In this study, we used human HCC tissue samples to verify that CHK1 is highly expressed in the tumor and associated with the advanced tumor stage. In addition, DNA damage induced by cisplatin or CHK1 inhibition augments cellular apoptosis through upregulating IRF1 in HCC cells. Furthermore, increased CHK1 suppresses IRF1 expression through proteolysis. Cisplatin and CHK1 inhibition upregulate MICA expression through IRF1 at the transcriptional level. The increased MICA expression is correlated with NK cell and CD8+T cell infiltration in human HCC tissues. These novel findings demonstrate that the interaction of CHK1 and IRF1 plays a key role in regulating the tumor immune microenvironment through MICA in HCC.

## 2. Materials and Methods

### 2.1. Patient Samples

Fifty patients with HCC and forty-six adjacent liver tissues were acquired from patients that underwent hepatectomy at the Department of General Surgery, The Second Affiliated Hospital of Guangxi Medical University (Nanning, China). The tissues were embedded in paraffin and made into sections for immunohistochemical (IHC) staining. In addition, 15 pairs of fresh HCC tumors and adjacent liver tissue specimens were obtained from patients who underwent hepatectomy or liver puncture. These tissues were used for subsequent qPCR experiments. All human tissues were obtained according to protocols (No. 2022-KY-0208) approved by the Second Affiliated Hospital Ethics Committee of Guangxi Medical University. Prior informed consent was obtained from the patients. 

### 2.2. Cell Line and Reagents

Hepatic cancer cell lines HepG2, Huh-7, and Hepa1-6 were obtained from the Chinese Academy of Sciences (Shanghai, China). All cells were cultured at 37 °C in a humidified incubator containing 5% CO_2_ using Dulbecco’s Modified Eagle Medium (DMEM) (Gibco, New York, NY, USA) supplemented with 100 U/mL penicillin, 100 μg/mL streptomycin (Invitrogen, Waltham, MA, USA) and 10% fetal bovine serum (FBS) (Sigma, Australia). The trypsin (Gibco, USA) was used during cell passage. In some experiments, the cells were treated with the CHK1 inhibitor prexasertib (LY2603618) purchased from MedChemExpress (Monmouth Junction, NJ, USA) or chemotherapeutic reagent cisplatin from APExBIO (Houston, TX, USA).

### 2.3. IHC Staining

Each collected tissue specimen was formalin-fixed and paraffin-embedded, and then the tumor and adjacent liver tissue were placed on the same block. Consecutive 5-micron thick sections were cut from each block and mounted on glass slides. After dewaxing, rehydration, and antigen repair, the anti-CHK1 (25887-1-AP), CD8 (66868-1-Ig), and CD56 (14255-1-AP) (Proteintech, Wuhan, China) anti-IRF1 (ab243895, abcam, Burlingame, CA, USA) antibodies were used respectively in IHC staining according to standardized institutional protocols. All slides were counterstained with hematoxylin, dehydrated, and sealed with neutral resin.

### 2.4. Lentivirus Infection

A lentiviral vector carrying human CHK1 cDNA, IRF1 cDNA, MICA cDNA, CHK1 shRNA, or empty vector was constructed by Genechem (Shanghai, China). According to the standardized protocols, Huh-7 cells were seeded and, the following day, infected with lentivirus for 48 h, according to the experimental MOI. Next, the culture medium was changed, and the cells stably expressing the target gene were sorted out by flow cytometry and expanded. The total RNA and cell lysate protein were extracted respectively for the following experiments.

### 2.5. Real-Time RT-PCR

Total RNA was isolated with TRIzol reagent and reversely transcribed to single-stranded with PrimeScript™ RT Master Mix (Takara Bio, Kyoto, Japan). Real-time quantitative polymerase chain reaction (PCR) was conducted using SYBR Premix Kit (Takara Bio, Kyoto, Japan). The reaction volume of the All-in-One qPCR Mix was 20 µL, including 1 µL of cDNA, 10 µL of 2× all-one qPCR Mix, 1 µL of 2 mmol/L reverse primer, 1 µL of 2 mmol/L forward primer, and 6µL of nuclease-free water. Primer sequences used were demonstrated in Appendix A. The relative changes of each gene were normalized to GAPDH mRNA and determined by the 2^−ΔΔCt^ method.

### 2.6. Western Blot

After the intervention, the cells were collected and lysed with high-efficiency RIPA (Solarbio, Beijing, China) for 30 min on ice and centrifuged at 12,000 R/min for 10 min at 4 °C. The supernatant was collected, and the protein content was detected by BCA Protein Assay Kit (P0012, Beyotime, Shanghai, China). The 30 μg protein solution volume was taken for sodium dodecyl sulfate and polyacrylamide gel electrophoresis. After being transferred to 0.45 µm polyvinylidene difluoride membrane, the membrane was sequestered with 5% BSA blocking solution for 1 h. The membrane was incubated with a 1:1000 dilution of primary antibodies overnight. Antibodies used for western blot were antibodies against CHK1 (25887-1-AP), MICA (12619-1-AP), and β-actin (20536-1-AP) (Proteintech, Wuhan, China); Anti-IRF1 (ab243895, abcam, Burlingame, CA, USA); anti-γ-H2AX (#80312) and GAPDH (CST, Danfoss, MA, USA). After being washed with Tris-buffered saline and Tween-20 (TBST) three times, the membranes were incubated with fluorescent secondary antibodies diluted 1:20,000 for 1 h and scanned by Odyssey (Odyssey CLX, LI-COR, Lincoln, NC, USA). The obtained WB target bands were quantified using Image J with the formula: Relative abundance (RA) = (Target band intensity)/(GAPDH intensity).

### 2.7. Immunofluorescent Staining and Flow Cytometry

Human Huh-7 cells were seeded in six-well plates for 24 h. After being treated with cisplatin for 24 h, cells were washed with PBS three times and fixed in 4% paraformaldehyde solution for 15 min. The cells were permeated with 0.5% Triton x-100 for 20 min at room temperature and incubated with primary MICA antibody (BS-0832R, BioSS, Beijing, China) at 4 °C overnight. Next, cells were incubated with Alexa Fluor 594 anti-rabbit antibody (CST, US) at 37 °C for 1 h. After being washed with PBS, slides were stained and capped with 4’, 6-diaminine-2’-phenylindole dihydrochloride (DAPI) and then observed using Olympus Fluview FV1000 III microscope (Olympus, Tokyo, Japan).

Huh-7 cells seeded in six-well plates were treated with cisplatin or dimethyl sulfoxide (DMSO) for 24 h. Then, 1 × 10^5^–3 × 10^5^ cells were collected, washed twice with pre-cooled PBS, and the supernatant was discarded after centrifugation. The Annexin v-apc /7 AAD apoptosis kit (AP105, Liankebio, Hangzhou, China) was used, and apoptotic cells were detected by flow cytometry (FACSCalibur, BD, Franklin, NJ, USA).

### 2.8. Co-Immunoprecipitation Assay

Co-Immunoprecipitation was performed according to the standardized protocol. The Immunoprecipitation Kit with Protein A+G Magnetic Beads was produced by Beyotime (P2179S, Beyotime, Shanghai, China). Huh-7 cells were treated with cisplatin with a dose of 10 µM for 24 h and washed with PBS three times, and then 100–200 μL inhibitor lysate was added for 10 min on ice. Lysates were centrifuged at 14,000× *g* at 4 °C for 5 min and incubated with an anti-IRF1 antibody (ab243895, abcam, Burlingame, CA, USA) or anti-CHK1 antibody (25887-1-AP, Proteintech, China) attached to Protein A+G magnetic beads. The protein samples were incubated with protein A+G magnetic beads bound with antibodies overnight at 4 °C on the shaker platform. Next, beads were collected by centrifugation at 3000× *g* for 5 min at 4 °C, washed in lysis buffer, and then resuspended in SDS gel loading buffer. The protein samples were prepared for western blot analysis.

### 2.9. Chromatin Immunoprecipitation (ChIP)

ChIP was conducted as previously described [14]. The ChIP-IT^®^ Express Enzymatic Magnetic Chromatin Immunoprecipitation Kit & Enzymatic Shearing Kit purchased from Active Motif (Carlsbad, CA, USA) was used for the ChIP assay. We designed seven pairs of qPCR primers in the human MICA transcript promoter region. Huh-7 cells were harvested after cisplatin was treated with a dose of 10 µM for 6 h. ChIP experimental procedures are performed according to the manufacturer’s instructions. Anti IRF1 antibody (ab243895, abcam, ChIP grade) was used in the experiment. The DNA collected by ChIP was detected by qPCR.

### 2.10. Single Cell RNA-Sequence Data Acquisition

In order to avoid the influence of different hepatitis backgrounds and tumor stages on the cellular components in TME, single cell RNA-sequence (scRNA-seq) data of two samples (GSM4505961 and GSM4505964) of GSE149614 were downloaded from the GEO database (https://www.ncbi.nlm.nih.gov/geo (accessed on 16 August 2022)). A total of 5615 cells and 25,712 genes from two patients (two tumor samples) were obtained. Relevant clinical information of the two patients is provided in A Single-Cell Atlas [23].

### 2.11. scRNA-Seq Data Preprocessing and Analysis

The R software “Seurat” package (v 4.2.0) was used to convert scRNA-seq data into Seurat objects. To retain high-quality scRNA-seq data, raw matrixes were filtered to remove cells (<200 transcripts/cell, >15% mitochondria genes) and genes (<3 cells/gene). Then, the “NormalizeData” function of the “Seurat package” was applied for data, and the normalization method was set as “LogNormalize”. Principal component analysis (PCA), together with JackStraw and ElbowPlot functions, was performed using the Seurat package in R software (v 4.2.1) to select the top 20 principal components (PCs). The “FindNeighbors” and “FindClusters” functions in the “Seurat package” were used for cell clustering analysis. Then, the “RunTSNE” and “RunUMAP” functions were used for cell clustering and visual analysis. The “FindAllMarkers” function was used to compare the differences in gene expression between a cluster and all other clusters. To identify the marker genes for each cluster, adjusted *p*-value < 0.05 and |log2 (fold change)| > 1 were used. Subsequently, cell cluster annotation was based on marker genes in the CellMarker databases [24] and genes reported in the literature.

### 2.12. Statistical Analysis

Statistical analyses were performed using GraphPad Prism 8 software (GraphPad Software Inc., San Diego, CA, USA). The *t*-test was used to test for statistical significance between different groups. For survival curves, the log-rank test was used to compare individual groups. Results were collective data from 2 to 4 repeat experiments. For all analyses, *p* < 0.05 was considered significant. Data were described by mean values ± standard deviation (SD) unless otherwise specified. 

## 3. Results

### 3.1. CHK1 in Human HCC Is an Oncogene to Predict Advanced Tumor Stage and Poor Prognosis

To explore CHK1 expression in the HCC tumor and background liver, we first analyzed CHK1 expression in the tumor and background liver tissue by IHC staining. Our data showed that CHK1 expression was positive in 46/50 (92%) of the HCC tumors and negative in 43/46 (93%) of the noncancerous liver (Figure 1a,b). The statistical analysis also showed that CHK1 expression was increased in the tumor compared to the background liver in these patients (Figure 1b).

In order to expand the sample size, we analyzed The Cancer Genome Atlas (TCGA) database and confirmed that CHK1 expression was increased in the tumor of patients with HCC (Figure 1c). Furthermore, CHK1 expression was significantly correlated with patients with higher BMI (>25), advanced tumor stage (T stage), and higher AFP level (>400 ng/mL) (Figure 1d–f). The survival analysis showed that higher CHK1 expression predicted a poorer prognosis (Figure 1g). Meanwhile, the ROC curve demonstrated that CHK1 served as an accurate biomarker for HCC prognosis (AUC = 0.951, Figure 1h).

### 3.2. CHK1 Inhibition Induced Cellular Apoptosis via DNA Damage in Human HCC Cells

As described previously, CHK1 inhibition induced apoptosis in HCC cells [15]. We then explored whether CHK1 inhibition caused apoptosis through DNA damage. Cisplatin is a common chemotherapeutic reagent used for patients diagnosed with HCC by causing cellular DNA damage. We first confirmed that cisplatin used in Huh-7 cells promoted γ-H2AX expression, a novel marker for DNA double-strand breaks (Figure 2a). Prexasertib is a known small-molecule inhibitor of CHK1. It decreased CHK1 expression in HepG2 and Huh-7 cells and increased γ-H2AX level (Figure 2b,c). Furthermore, the flow cytometry assay verified significantly increased apoptosis in Huh-7 cells treated with prexasertib (Figure 2d,e).

### 3.3. CHK1 Directly Binds IRF1 and Exerts Proteolytic Effect in HCC Cells Induced by DNA Damage

Previously we showed that IRF1 downregulated CHK1 expression via miR-195 [15]. However, another study found that CHK1 promoted substrate proteolysis in HCC cells [16]. Thus, we wondered whether CHK1 exerted a proteolytic effect on IRF1. We first confirmed that CHK1 inhibition had no effect on IRF1 mRNA expression in Huh-7 cells (Figure 3a). The decreased CHK1 expression in Huh-7 cells induced by molecular inhibitor prexasertib upregulated IRF1 expression (Figure 3b). Meanwhile, downregulated CHK1 expression in Huh-7 cells infected by lentiviral CHK1 shRNA promoted IRF1 expression (Figure 3c,d). Conversely, exogenous expression of CHK1 by lentiviral CHK1 transduction downregulated IRF1 expression in HCC cells (Figure 3e,f). Interestingly, the decreased IRF1 level was reversed by bortezomib, a proteasome inhibitor (Figure 3f). 

Next, we investigated whether a direct interaction existed between CHK1 and IRF1 in HCC cells under the condition of DDR. The co-immunoprecipitation (co-IP) assay was performed to determine whether CHK1 could directly bind IRF1 protein in the Huh-7 cells induced by cisplatin. Co-IP pull-down assay showed that antibody to CHK1 bound IRF1 (Figure 3g). Control IgG antibody did not pull down IRF1. CHK1 expression was confirmed by western blot (Figure 3g). Additionally, antibodies to IRF1 (but not IgG) pulled down CHK1 (Figure 3h). IRF1 expression was confirmed by western blot (Figure 3h). Taken together, CHK1 suppressed IRF1 through binding IRF1 and exerting proteolysis in HCC cells under the condition of DNA damage. 

### 3.4. DNA Damage Induced MICA Expression via IRF1 in HCC Cells

The previous study showed that NKG2D ligands were induced by DNA damage [17,18]. The NKG2D ligands mediated the recruitment and activation of NK cells and CD8+T cells through NKG2D receptors [19,20]. We first confirmed that cisplatin and prexasertib upregulated NKG2D ligands mRNA expression, remarkably MICA mRNA level (Figure 4a,b). Furthermore, immunofluorescent staining and western blot for MICA protein expression showed upregulated levels in Huh-7 cells induced by cisplatin (Figure 4c–e). The increased MICA expression was confirmed in HepG2 cells induced by prexasertib (Figure 4f).

CHK1 inhibition and cisplatin upregulated IRF1 expression (Figure 3b,d–f). In addition, our previous study found that IRF1 functioned as a central transcriptional factor in regulating immune cell infiltration in HCC [14,15]. Thus, we explored the correlation between IRF1 and MICA. We first labeled different types of cells in the human HCC tumor with their specific marker according to scRNA-seq in the GEO database (Figure 4g). As we expected, IRF1 positive expression in tumor cells was consistent with MICA positive (Figure 4h). Meanwhile, a significantly higher IRF1 level was found in the HCC tumor cells with positive MICA expression than those with negative (Figure 4i). Thus, scRNA-seq analyses confirmed that MICA expression positively correlated with IRF1 in HCC cells. 

To investigate whether IRF1 was an upstream regulator of MICA, we upregulated IRF1 expression in human HCC cells through lentiviral IRF1 transduction. As we expected, the MICA protein level was markedly increased in both Huh-7 cells and HepG2 cells (Figure 5a,b). Conversely, the increased MICA expression induced by cisplatin was attenuated when endogenous IRF1 expression was downregulated by IRF1 siRNA in human Huh-7 cells (Figure 5c). These findings were consistent with the notion that DNA damage promoted MICA expression via IRF1 in HCC cells.

### 3.5. IRF1 Upregulates MICA at the Transcriptional Level in HCC Cells

Next, we investigated whether IRF1 regulates MICA expression at the transcriptional level. We analyzed two kilobases in the 5′-upstream flanking region of the human MICA gene using PROMO bioinformatics software. We identified seven putative IRF1 binding elements (B1-B7) in the human MICA promoter (Figure 5d). 

To discover whether IRF1 binds directly to the MICA promoter and whether those binding elements are functionally active, we performed chromatin immunoprecipitation (ChIP) in the cell lysis of cisplatin-treated Huh-7 cells with anti-IRF1 antibody and PCR primers spanning each element. Cisplatin-induced IRF1 transcriptional activity was shown with anti-IRF1 antibodies (but not control IgG antibody) for binding at the B6 response element at −1756 base pair in the 5′-flanking region of the MICA promoter (Figure 5e). However, no transcriptional binding was found at other putative elements.

### 3.6. MICA Expression Is Positively Correlated with NK Cells and CD8+T Cells Infiltration in Human HCC

Since the NKG2D receptor was expressed by activated NK cells and CD8+T cells in humans, we next investigated the correlation of MICA expression with NK cells and CD8+T cell infiltration in human HCC. We verified a significant positive correlation between MICA & CD56 (a marker of NK cell) and MICA & CD8 mRNA expression in tumors and background liver from patients diagnosed with HCC (Figure 6a,b). Furthermore, we analyzed the correlation of MICA expression and NK cells & CD8+T cells infiltration in HCC tumor from the TIMER database (Figure 6c,d). Meanwhile, IHC staining for MICA expression decreased in the tumor compared to the background liver (Figure 6e,f). The significant positive correlation of MICA expression and NK cells & CD8+T cell infiltration in HCC tumor and non-cancerous tissue was confirmed by IHC staining (Figure 6g,h). Collectively, our results suggested that MICA expression may increase NK cell and CD8+T cell infiltration in HCC.

A summary figure illustrates that DDR inducers (e.g., chemotherapy agents) regulated the interaction of CHK1 and IRF1 to promote MICA transcription via IRF1 in HCC cells. The increased MICA expression in the cellular membrane may recruit NK cells and CD8+T cell infiltration in HCC (Figure 6i). 

## 4. Discussion

The major findings of this study are: (1) IHC staining for CHK1 was increased in human HCC tumors compared to background liver, and high CHK1 expression predicted advanced tumor stage and poor prognosis. (2) Cisplatin and CHK1 inhibition caused DNA damage to promote apoptosis in HCC cells. (3) The increased CHK1 suppressed IRF1 expression, and CHK1 directly bound IRF1 to exert proteolytic effect on IRF1. (4) DNA damage induced MICA expression via IRF1 in HCC cells, and the novel molecular mechanism identified was trans-acting IRF1 binding to a specific cis-acting IRF response element at −1756 bp in the MICA promoter region to induce MICA gene transcription. (5) The MICA expression positively correlated to NK cells and CD8+T cell infiltration in human HCC.

CHK1 has been regarded as an oncogene and increased in numerous human malignant tumors, including breast cancer, colon cancer, liver cancer, gastric cancer, and acute lymphoblastic leukemia [25]. We verified high CHK1 expression in human HCC tumor tissue by IHC staining. The statistical analyses on the TCGA database also confirmed that CHK1 positively correlated with tumor grade and disease prognosis. Meanwhile, the ROC curve verified CHK1 as an accurate biomarker for HCC prognosis. Importantly, a high positive CHK1 expression rate (92%) in HCC tumors serves as a potential negative biomarker for HCC correlating with greater T stage, high AFP levels, and worse overall survival. Hence, the identification of high CHK1 expression in HCC tumors may potentially be used to identify aggressive HCC, which could impact clinical treatment recommendations.

In this study, we investigated DNA damage in HCC cells induced by CHK1 inhibitor prexasertib or the chemotherapeutical drug cisplatin. When DNA damage occurs, and double-stranded breaks (DSBs) quickly form, it is always followed by phosphorylation of the H2AX, a variant of the H2A histone family [26]. This newly phosphorylated protein, γ-H2AX, is the first step in recruiting and localizing DNA repair proteins, and then γ-H2AX foci form. Thus, γ-H2AX foci can represent DSBs and become a novel biomarker for DNA damage [27]. Furthermore, failure of DNA double-stranded breaks repair induced cellular apoptosis [28]. We found a remarkably increased γ-H2AX level in human HCC cells induced by cisplatin or CHK1 inhibition. Meanwhile, the increased cellular apoptotic rate in HCC cells treated with CHK1 inhibition was detected by flow cytometry. Collectively, our findings were consistent with the notion that CHK1 inhibition and cisplatin induce DNA damage to promote apoptosis in HCC cells. 

Our previous study showed that IRF1 and CHK1 cooperate to modulate apoptotic signaling pathways in HCC [15]. In this study, we found that decreased IRF1 expression induced by upregulated CHK1 was reversed by bortezomib, a highly effective selective and reversible proteasome inhibitor [29]. Meanwhile, CHK1 inhibition upregulated IRF1 expression, and Co-IP verified CHK1 binding IRF1 in HCC cells under the condition of DNA damage response caused by cisplatin. This evidence confirmed that CHK1 bound IRF1 to exert a proteolytic effect on IRF1. Additionally, our previous study observed low IRF1 expression in HCC tumors compared to background liver [30]. The high CHK1 level in the tumor may contribute to low IRF1 expression. 

In this study, we explained the molecular mechanism of DNA damage in HCC cells induced by CHK1 inhibitors or cisplatin recruiting anti-tumor immune cells, including NK cells and CD8+T cells. We found that CHK1 inhibition and cisplatin upregulated MICA expression in HCC cells. These findings confirmed results from previously published reports [17,31]. The NKG2D system is an effective cancer immune surveillance mechanism composed of NKG2D receptors and their homologous ligands, which stimulate the expression of NKG2D in immune cells under various stress signals, including infection, DNA damage, heat shock, and hyperplasia [31,32]. 

Importantly, our novel finding is that DNA damage induced by CHK1 inhibition and cisplatin upregulated MICA expression through IRF1 binding to a specific cis-acting response element promoting MICA gene transcription. 

According to our studies, this CHK1/IRF1/MICA axis activates anti-tumor immune cells to enhance tumor disruption through the interaction of miR-195. The miR-195 may be a potential integrated factor of these signaling pathways and a therapeutic target in human liver cancer. Thus, more experimental evidence will be needed to verify miR-195 directly suppressing CHK1 to upregulate MICA expression and attract immune cells by using the murine HCC model and analyzing the correlation of miR-195 with MICA in human HCC tissues.

In addition, we found that MICA expression positively correlated with NK cells and CD8+T cell infiltration in HCC. These results confirmed our previous study that IRF1 regulated immune cell infiltration in murine HCC [14]. Thus, the DDR pathway communicates with IRF1-MICA signaling to mediate the recruitment of NK cells and CD8+T cells. Meanwhile, since our previous study also showed that IRF1 recruited NK cells and CD8+T cells through CXCL10/CXCR3 axis [14], we investigated whether IRF1-MICA signaling recruited anti-tumor immune cells via the CXCR3 receptor. However, the analysis of the correlation of MICA and CXCR3 through protein interaction network (PPI) showed no direct binding of MICA to CXCR3, and whether NKG2D or other unknown receptors expressed on NK cells and CD8+T cells playing this role needed further studies.

Although we found MICA positively correlated with anti-tumor immune cell infiltration in HCC, there were different results about the correlation of NKG2D ligands with tumor aggressiveness and the clinical prognosis in different cancer types and individuals [21,33]. Some studies indicated that shedding of MICA or other NKG2D ligands contributed to tumor immune escape by releasing soluble MICA (sMICA) molecules binding to NKG2D receptors and inducing its down-modulation and degradation [34]. NK cell and dendritic cell functions were impaired by the sMICA in advanced human HCC [35]. Thus, further studies are needed to explore the correlation mechanism of MICA with HCC aggressiveness involving the increased sMICA level.

In addition, the combination of a DNA damage repair inhibitor with an immune checkpoint inhibitor is ongoing in clinical trials for some solid tumors, including breast cancer, ovarian cancer, and pancreas cancer, based on the preclinical evidence of efficacy and no obvious overlapping toxicity [36,37]. Our findings of DNA damage induced by DNA damage repair blockage promoting recruitment of anti-tumor immune cells provide the preclinical evidence for the combination of CHK1 blockade and ICI use in HCC. Further studies will be needed to confirm the efficacy and decrease the associated adverse reaction of combined therapy. 

In summary, our study explored the mechanisms of DDR signaling communicating with the IRF1 pathway in regulating the HCC tumor microenvironment and can potentially serve as a therapeutic strategy in treating patients with advanced HCC. 

## 5. Conclusions

DNA damage regulates the interaction of CHK1 and IRF1 to activate anti-tumor immunity via the IRF1-MICA pathway in HCC.

## Figures and Tables

**Figure 1 cancers-15-00850-f001:**
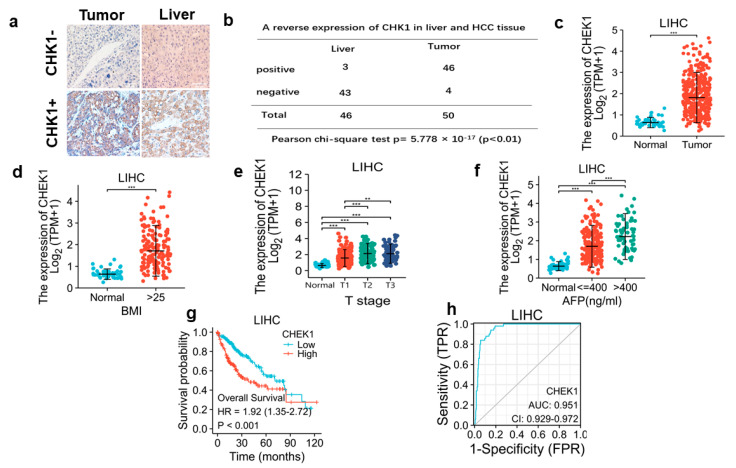
CHK1 is upregulated in HCC tumor and predict advanced tumor stage and poor prognosis. *(***a**) IHC staining for CHK1 in HCC tumor (*n* = 50) and background liver tissue (*n* = 46). The representative IHC images (×200 magnification) are shown for CHK1 protein expression (brown) in tumor and liver tissue. (**b**) Tissue sections of the HCC tumor and liver were IHC stained for CHK1 expression, and their difference was analyzed by Pearson chi-square test (*p* < 0.01). Statistical analyses on the CHEK1 mRNA expression in normal liver and HCC tumor (**c**), in patients with normal and high BMI (**d**), in patients with different tumor stages (**e**), and in patients with different AFP levels (**f**) are shown. *(***g**) The clinical significance of CHEK1 mRNA expression in HCC overall survival is evaluated through Kaplan-Meier analysis. *(***h**) The sensitivity of CHEK1 mRNA expression predicting the prognosis of HCC patients is evaluated by the ROC curve. ( ** *p* < 0.01, *** *p* < 0.001).

**Figure 2 cancers-15-00850-f002:**
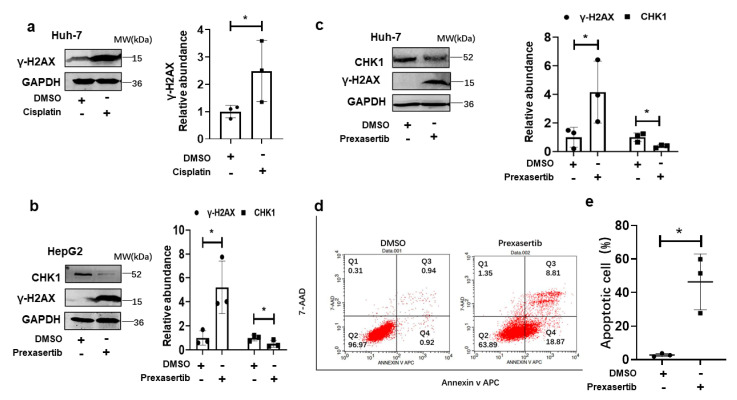
CHK1 inhibition induces DNA damage to cause apoptosis in HCC cells. (**a**) Western blot and quantitative analysis for γ-H2AX level is shown in Huh-7 cells induced by cisplatin (5 µM) for 24 h. The γ-H2AX expression is measured by western blot in HepG2 (**b**) and Huh-7 cells (**c**) induced by prexasertib with a dose of 5 µM for 24 h. (**d**) Representative image of FACS analysis of apoptotic Huh-7 cell rate treated with DMSO or prexasertib with a dose of 5 µM for 24 h are shown. (**e**) The statistical summary of apoptotic Huh-7 cell rate is shown with *t* test (*n* = 3). Data represent mean ± SD, * *p* < 0.05. Western blot image shown are representative of 3 experiments. FACS assay shown are representative of 3 independent experiments. The uncropped blots are shown in Appendix A.

**Figure 3 cancers-15-00850-f003:**
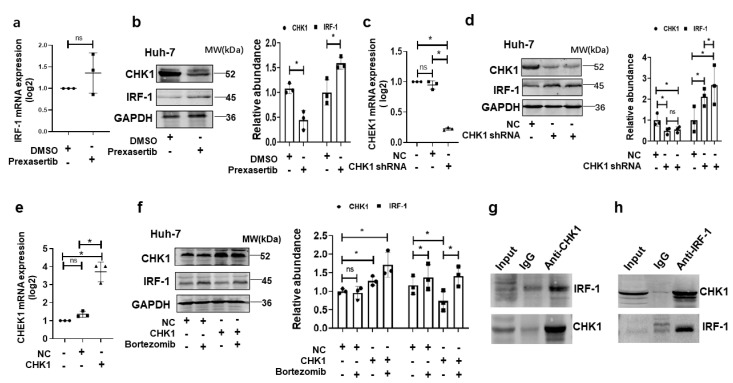
CHK1 exerts proteolytic effects on IRF1. (**a**) IRF1 mRNA expression is detected by qPCR in Huh-7 cells treated with prexasertib with a dose of 5 µM for 24 h. (**b**) The whole lysate for IRF1 and CHK1 protein expression are measured by western blot in Huh-7 cells induced by prexasertib. The quantitative analyses for CHK1 and IRF1 are shown on the right. (**c**) CHEK1 mRNA expression is measured by qPCR in Huh-7 cells infected by lentiviral CHK1 shRNA or empty vector (NC) at a dose of 50 MOI for 48 h. (**d**) CHK1 and IRF1 proteins are detected by western blot in Huh-7 cells infected by lentiviral CHK1 shRNA or empty vector (NC). (**e**) IRF1 mRNA expression is measured by qPCR in Huh-7 cells infected by lentiviral CHK1 cDNA or empty vector (NC) at a dose of 50 MOI for 48 h. (**f**) CHK1 and IRF1 protein expression are measured by western blot in Huh-7 cells induced by overexpressed CHK1 (CHK1) or negative control (NC). The decreased IRF1 protein level induced by overexpressed CHK1 is reversed by bortezomib with a dose of 250 nM for 24 h. (**g**) Co-IP assay of Huh-7 cells induced by cisplatin with a dose of 10 µM for 24 h. The interaction of IRF1 with CHK1 is determined by pulling down with anti-CHK1 antibodies and western blot with anti-IRF1 antibodies. (**h**) Anti-IRF1 antibody and western blot with anti-CHK1 antibody shows IRF1 pulled down CHK1. Each point represents an independent experiment. Data represent mean ± SD, * *p* < 0.05, ns not significant. Western blot image shown are representative of 3 experiments. The uncropped blots are shown in Appendix A.

**Figure 4 cancers-15-00850-f004:**
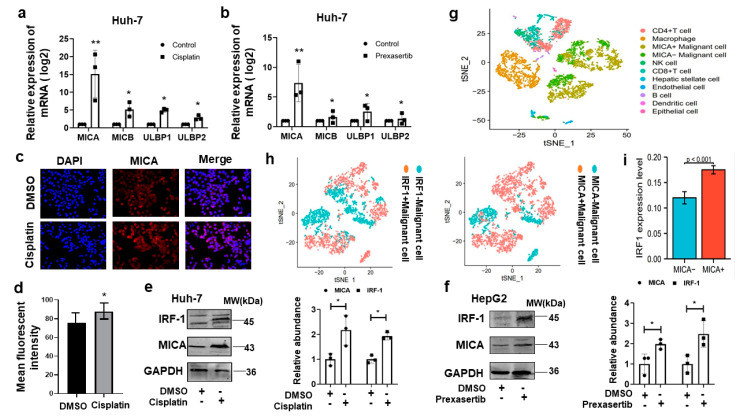
DNA damage induced MICA expression via IRF1 in HCC cells. The relative mRNA expression of NKG2D ligands is determined by qPCR in Huh-7 cells induced by cisplatin with a dose of 5 µM for 24 h (**a**) and prexasertib with a dose of 5 µM for 24 h (**b**). (**c**) The representative immunofluorescent images (×400 magnification) show MICA protein level (red staining) in Huh-7 cells induced by cisplatin with a dose of 5 µM for 24 h. (**d**) Quantification of MICA by mean fluorescent intensity is shown. The representative western blot image detected for IRF1 and MICA expression in Huh-7 (**e**) and HepG2 cells (**f**) induced by cisplatin or prexasertib with a dose of 5 µM for 24 h are shown, respectively. The quantitative analyses are shown on the right. (**g**) The different types of cells in the human HCC tumor are shown by their own specific marker on scRNA-seq from the GEO database. (**h**) IRF1 expression is consistent with MICA in tumor cells. (**i**) The statistical analysis of IRF1 expression level in HCC tumor cells with positive MICA expression is shown. Data represent mean ± SD, * *p* < 0.05, ** *p* < 0.01. The western blot and immunofluorescent results are representative images from three independent experiments. The uncropped blots are shown in Appendix A.

**Figure 5 cancers-15-00850-f005:**
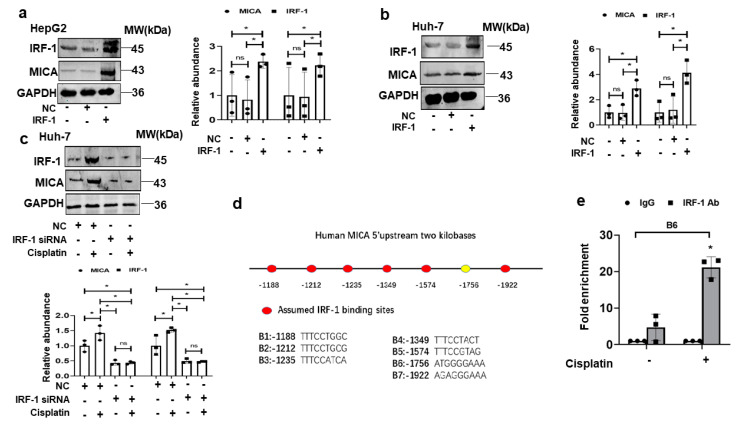
IRF1 promotes MICA transcription in response to DNA damage. The whole lysate for IRF1 and MICA protein determined by western blot is shown in HepG2 cell (**a**) and Huh-7 cell (**b**) infected by lentiviral IRF1 cDNA (50 MOI) for 48 h. The quantitative analyses are shown on the right. (**c**) The western blot for MICA and IRF1 protein expression was shown in Huh-7 cells induced by cisplatin at a dose of 10 uM with or without IRF1 siRNA transfection for 48 h. (**d**) The schematic representation of IRF1 binding sites in the human MICA promoter region as predicted by PROMO bioinformatics software are shown. (**e**) ChIP assay is performed with IgG or anti IRF1 antibody in Huh-7 cells, which are induced by cisplatin (10 uM) for 24 h. The qPCR analyses of immunoprecipitated DNA are conducted using the primers, designed to amplify the indicated region of the MICA promoter. Data represent the mean ± SD, * *p* < 0.05, ns not significant. The uncropped blots are shown in Appendix A.

**Figure 6 cancers-15-00850-f006:**
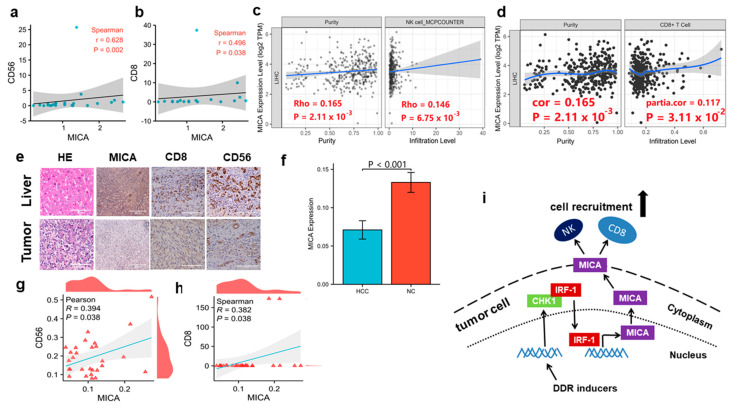
The correlation between MICA and immune cell infiltration in HCC. (**a**,**b**) The correlation of MICA & CD56 and MICA & CD8 mRNA expression determined by qPCR in pairs of tumor and noncancerous liver tissues from HCC patients were shown (*n* = 15). (**c**,**d**) The correlation of MICA expression and NK cells & CD8+T cells infiltration in HCC tumors from the TIMER database is shown. (**e**) The representative IHC images for MICA, CD8, and CD56 in tumor and noncancerous liver are shown. (**f**) The statistical analysis of MICA protein expression in the tumor compared to the background liver is shown. (**g**,**h**) Statistical IHC scores for MICA & CD56 and MICA & CD8 are used to determine the relationship in MICA, NK cells, and CD8+T cells (*n* = 15). (**i**) A carton depicts that DDR inducers regulate the interaction of CHK1 (green) and IRF1 (red) to promote MICA transcription via IRF1. The increased MICA (purple) activates NK cells (NK) and CD8+T cells (CD8) recruitment in HCC. Each data point represents one patient.

## Data Availability

All generated, as well as analyzed, data in the present research are contained in this manuscript.

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
