# Peer review of "Inhibition of Checkpoint Kinase 1 (CHK1) Upregulates Interferon Regulatory Factor 1 (IRF1) to Promote Apoptosis and Activate Anti-Tumor Immunity via MICA in Hepatocellular Carcinoma (HCC)"

_cancers, 2023, doi:10.3390/cancers15030850_

Round 1

Reviewer 1 Report

References

The manuscript “Inhibition of Checkpoint kinase 1CHK1upregulates Interferon regulatory factor 1 (IRF1) to promote apoptosis and activate anti-tumor immunity via MICA in hepatocellular carcinoma” by Xicai Li et al describing the mechanistic aspects of DDR signaling /IRF1 pathway in the regulation of HCC tumor microenvironment. By using single cell RNA sequencing analysis authors have demonstrated that that MICA expression is positively correlated with IRF1 in HCC cells. Cisplatin and CHK1 inhibition upregulate MICA expression through IRF1 transcription. MICA enhances NK cell and CD8+ T cell infiltration into tumor, leading to activation of antitumor activity in HCC.  Authors underscore the importance e of this pathway from the perspective of a potential therapeutic target in the treatment of advanced HCC patients. Nonetheless, it would be interesting if authors could demonstrate the upregulation of expression levels of CXCR3 on CD8+ T cells along with MICA/IRF1 activation. Please see the references below.

https://www.nature.com/articles/s41416-021-01337-6

https://pubmed.ncbi.nlm.nih.gov/33689775/

Y Yan et al 2021

Author Response

Reviewer #1:

The manuscript “Inhibition of Checkpoint kinase 1(CHK1)upregulates Interferon regulatory factor 1 (IRF1) to promote apoptosis and activate anti-tumor immunity via MICA in hepatocellular carcinoma” by Xicai Li et al describing the mechanistic aspects of DDR signaling /IRF1 pathway in the regulation of HCC tumor microenvironment. By using single cell RNA sequencing analysis authors have demonstrated that that MICA expression is positively correlated with IRF1 in HCC cells. Cisplatin and CHK1 inhibition upregulate MICA expression through IRF1 transcription. MICA enhances NK cell and CD8+ T cell infiltration into tumor, leading to activation of antitumor activity in HCC.  Authors underscore the importance of this pathway from the perspective of a potential therapeutic target in the treatment of advanced HCC patients. Nonetheless, it would be interesting if authors could demonstrate the upregulation of expression levels of CXCR3 on CD8+ T cells along with MICA/IRF1 activation. Please see the references below.

https://www.nature.com/articles/s41416-021-01337-6

https://pubmed.ncbi.nlm.nih.gov/33689775/

Y Yan et al 2021

This is novel idea. However, MICA is the ligand of NKG2D receptor which mainly expresses on CD8+T cells and NK cells. CXCR3 is a receptor expressed on NK cells and CD8+T cells according to our previously published paper. We analyzed the relationship between MICA and CXCR3 through protein protein interaction network (PPI) and found no direct correlation of MICA and CXCR3. It looks like that IRF1 may play diverse roles as a central transcriptional factor in different signaling pathways including IFN pathway. We explained these findings by expanding the discussion.

Reviewer 2 Report

Please find below my comments on the manuscript entitled: “Inhibition of Checkpoint kinase 1 (CHK1) upregulates Interferon regulatory factor 1 (IRF1) to promote apoptosis and activate anti-tumor immunity via MICA in hepatocellular carcinoma (HCC)”, written by “Xicai Li, Jingquan Huang, Qiulin Wu, Qiang Du, Yingyu Wang, Yubin Huang, Xiaoyong Cai, David A. Geller, and Yihe Yan”, that was submitted for consideration to Cancers.

In the last few years targeting the DNA damage response, of which CHK1 is a key component, has emerged as a new therapeutic approach that shows great promise. The current study aims to shed some light into the mechanism regulating CHK1 signaling within the TME in HCC. The main finding of this study is that CHK1, which is upregulated in HCC, binds to IRF1 leading to its proteolysis. The authors show that CHK1 inhibition leads to IRF1 up-regulation with resulting promotion of cell death and potential homing of immune cells into the tumor.

The experiments included in this study are well introduced, logically designed, and the conclusions are supported by the included data. However, this is the third study published by this group of authors on the IRF1/CHK1 axis, and I found that some of the data included in the submitted study (e.g., up-regulation of CHK1 in HCC), are presented as "new," whereas the identical findings had already been reported in their previous manuscript, albeit using different methodologies to detect them. In addition, the discussion should be expanded to include the results of previously published studies on IRF1/CHK1. I am confident that these changes will increase the impact of their study.

Therefore, for the manuscript to be suitable for publication in Cancers, I ask the authors to include the changes indicated in the comments below.

Comments:

The authors have previously shown that CHK1 is increased in HCC and is associated with poor prognosis, so this part should be removed from the manuscript and refer to it only. The IHC data can remain as independent confirmation of your previous study.

The authors have also shown previously that the CHK1/IRF1 axis, through the interaction of miR-195, could be the driving force responsible for the attraction of immune cells in tumor disruption, please discuss.

Overall, the current data seem to integrate well with your previous work on CHK1/IRF1/miR-195. Please discuss the potential integration of these pathways and their potential in the treatment of liver cancer in humans.

Author Response

Reviewer #2:

  1. The authors have previously shown that CHK1 is increased in HCC and is associated with poor prognosis, so this part should be removed from the manuscript and refer to it only. The IHC data can remain as independent confirmation of your previous study.

We thank for these suggestions. Actually, we analyzed TCGA database and confirmed that CHK1 expression was increased in the tumor of patients with HCC (Fig. 1c). Also, CHK1 expression was significantly correlated with patients with higher BMI, advanced tumor stage, and higher AFP level (Fig. 1d-1f). Furthermore, we used ROC curve to confirm CHK1 as an accurate biomarker for HCC prognosis (Fig. 1h). These data represent new findings and were not reported in the previous paper (Ref. 15).  Previously, we only showed CHEK1 mRNA expression from HCC patients (n=30) and CHK1 protein level using immunofluorescent staining and western blot from HCC patients (n=6). Thus, we verified that increased CHK1 expression in HCC was associated with poor prognosis using a larger cohort of patients and additional prognostic markers to provide more comprehensive data than those in the previous study.

  1. The authors have also shown previously that the CHK1/IRF1 axis, through the interaction of miR-195, could be the driving force responsible for the attraction of immune cells in tumor disruption, please discuss.

Overall, the current data seem to integrate well with your previous work on CHK1/IRF1/miR-195. Please discuss the potential integration of these pathways and their potential in the treatment of liver cancer in humans.

We thank for this suggestion. We expanded the discussion to reflect this point.

Reviewer 3 Report

The manuscript entitled “Inhibition of Checkpoint kinase 1 (CHK1) upregulates Interferon regulatory factor 1 (IRF1) to promote apoptosis and activate anti-tumor immunity via MICA in hepatocellular carcinoma (HCC)” reports that DNA damage activates antitumor immunity in HCC by regulating the IRF1-MICA pathway through the interaction of CHK1 and IRF1. However, the following points must be addressed before the manuscript can be suitable for publication.

Comments:

1.       The authors need to provide more information on TME and MICA in the introduction.

2.       The authors should analyze the correlation between CHK1 and IRF1 using the TCGA database.

3.       Why did most experiments use Huh7 cells rather than HepG2 cells? Does HepG2 cells produce the same results as Huh7?

4.       The western blot images in Figures 2, 3, 4, and 5 should be additionally presented as quantitative graphs with statistical analysis.

5.       In Figure 4e, the expression level of MICA by cisplatin was checked. Additionally, the level of MICA expression by prexasertib should be investigated.

6.       In Figure 6e, scale bars should be indicated.

Author Response

Reviewer #3:

The manuscript entitled “Inhibition of Checkpoint kinase 1 (CHK1) upregulates Interferon regulatory factor 1 (IRF1) to promote apoptosis and activate anti-tumor immunity via MICA in hepatocellular carcinoma (HCC)” reports that DNA damage activates antitumor immunity in HCC by regulating the IRF1-MICA pathway through the interaction of CHK1 and IRF1. However, the following points must be addressed before the manuscript can be suitable for publication.

  1. The authors need to provide more information on TME and MICA in the introduction.

We thank for the suggestion. We provided more information on MICA and TME in the introduction.

  1. The authors should analyze the correlation between CHK1 and IRF1 using the TCGA database.

We analyzed the correlation between CHK1 and IRF1 in the TCGA database through cBioPortal (Spearman: 0.08, P=0.130). The TCGA database provides the interaction of different gene mRNA expression level.  In our current study, CHK1 suppressed IRF1 expression through proteolysis. Conversely, IRF1 decreased CHK1 protein level via microRNA-195 at post-transcriptional level according to our previous published paper. Thus, the interaction of CHKI and IRF1 existed.

  1. Why did most experiments use Huh7 cells rather than HepG2 cells? Does HepG2 cells produce the same results as Huh7?

Actually, we used HepG2 cells and acquired the same results as Huh7 cells. We demonstrated the typical images according to WB, PCR and IF experiments. We also provided some typical images from HepG2 cells in Fig. 2b, Fig. 4f and Fig. 5a.

  1. The western blot images in Figures 2, 3, 4, and 5 should be additionally presented as quantitative graphs with statistical analysis.

We thank for this suggestion.  We provided quantitative graphs with statistical analyses for these original WB images and showed in the right of Fig. 2a-2c, 3b, 3d, 3f, 4e, 4f, 5a, 5b, and in the bottom of Fig. 5c.

  1. In Figure 4e, the expression level of MICA by cisplatin was checked. Additionally, the level of MICA expression by prexasertib should be investigated.

Thank you for this recommendation. We performed additional experiments and show increased MICA expression along with IRF-1 induced by prexasertib (shown in new Figure 4f).

  1. In Figure 6e, scale bars should be indicated.

We thank for this suggestion. We showed the scale bars in the new Fig. 6e.

Round 2

Reviewer 3 Report

The authors properly revised the article.